

# A novel incomplete hesitant fuzzy information supplement and clustering method for large-scale group decision-making

Jingdong Wang[1], Wenhui Wang[1], Fanqi Meng[1,2], Peifang Wang[1], Xuesong Wang[1], Shuang Wei[1], Tong Liu[1] and Shuaisong Yang[1]

[1] School of Computer Science, Northeast Electric Power University, Jilin, Jilin, China
[2] School of Information Engineering, Guangdong Atv Academy For Performing Arts, Dongguan, Guangdong, China

Corresponding authors
Fanqi Meng,
mengfanqi@neepu.edu.cn
Peifang Wang,
2202000657@neepu.edu.cn

## ABSTRACT

Clustering is an effective means to reduce the scaling of large-scale group decision-making (LSGDM). However, there are many problems with clustering methods, such as incomplete or ambiguous information usually provided by different decision makers. Traditional clustering methods may not be able to handle these situations effectively, resulting in incomplete decision-making information. Calculating the clustering centers may become very complex and time-consuming. Inappropriate distance weights may also lead to incorrect cluster assignments, and these problems will seriously affect the clustering results. This research provides a novel incomplete hesitant fuzzy information supplement and clustering approach for large-scale group decision-making in order to address the aforementioned difficulties. First, the approach takes into account the trust degradation and the inhibition of relationships of distrust in the process of trust propagation, and then it builds a global and local network of trust. A novel supplemental formula is provided that takes into account the decision-preference maker's as well as the trust-neighbor's information, allowing the decision-neighbor maker's recommendation to be realized. Therefore, an improved distance function can be proposed to calculate the weights by combining the relative standard deviation theory and selecting the selected clustering centers by using the density peaks in order to optimize the selection of clustering centers and reduce the complexity and scaling of the decision. Finally, an example is presented to demonstrate how the proposed method can be applied. The consistency index and comparison experiments are used to evaluate if the suggested approach is effective and reliable.

# INTRODUCTION

Decision-making is a highly interdisciplinary field of study that is essentially a process of value discovery and judgment that is a choice between two or more alternatives that make an irrevocable allocation of resources. Due to the multi-objective, uncertainty problems are

multi-objective, uncertain, and dynamic, and there are limitations in individual knowledge, experience, and information, the ability of a single decision maker is overstretched. The ability of a single decision maker is overstretched, so it is necessary to rely on the wisdom of the group. Group decision-making (GDM) is the process of assembling and analyzing the preferences of group members, so as to obtain a satisfactory consensus solution. However, with the rapid development of Internet technology and social media, decision-making problems involve more and more decision-makers with different backgrounds, such as government, business, and academia, who have complex social relationships and frequent communication with each other. The traditional group decision-making theory is no longer sufficient to solve such problems, so the study of large-scale group decision making (LSGDM) has gradually attracted attention.

Our study is based on a typical large-scale group decision-making case that originated in the field of supply chain management. Specifically, this is a decision problem involving a large manufacturing company that needs to select the best supplier from among multiple potential suppliers to fulfill its raw material needs. The case involved multiple decision makers, including production, purchasing, quality control, and senior management. Each decision maker has different weights and preferences, and each supplier has different characteristics. The case provides a typical large-scale group decision-making situation that reflects the complexity of the real world. The purpose of this study is to bring a novel approach to the field of LSGDM, aiming to increase the effectiveness of information supplementation and reduce the computational complexity, while improving the quality of clustering results. With this approach, we will be able to better cope with the information incompleteness and complexity in large-scale group decision making and provide decision makers with more feasible decision support. In the subsequent parts of this article, we will present the principles, applications, and evaluation results of the method in detail to demonstrate its effectiveness and reliability in LSGDM. Although scholars have begun researching methods for supplementing and clustering large-scale group decision-making with incomplete hesitant fuzzy information, challenges persist. Firstly, in constructing trust networks, only partial information utilization is considered, and trust relationships are assumed to be symmetrical. The inhibitory effect of distrust relationships is neglected, leading to trust information supplementation deviating from reality and diminishing decision quality. Secondly, existing studies often overlook situations where preferences among decision-makers reflect degrees of similarity, neglecting the significance of supplementing missing decision information. Lastly, clustering methods based on the FCM algorithm, commonly used in the research, require pre-determination of cluster centers, making results susceptible to center selection and prone to issues such as information loss, instability, and susceptibility to noise interference.

The purpose of this study is to bring a novel approach to the field of LSGDM, aiming to increase the effectiveness of information supplementation and reduce the computational complexity, while improving the quality of clustering results. With this approach, we will be able to better cope with the information incompleteness and complexity in large-scale group decision making and provide decision makers with more feasible decision support. In the subsequent parts of this article, we will present the principles, applications, and

evaluation results of the method in detail to demonstrate its effectiveness and reliability in LSGDM.

The remaining part of the article is organized as follows. "Literature Review" summarizes and analyzes existing literature on the incomplete preference information problem in large-scale group decision-making. It highlights key viewpoints and methods while pointing out persisting challenges. "Preliminaries" provides the related concepts of hesitant fuzzy sets (HFSs), the social network, and some method-related definitions. "The Proposed Method for the LSGDM Clustering Problem" constructs the model of Supplement and clustering, mainly including trust network establishment, decision information supplement method, and weighted hesitant fuzzy clustering method based on relative standard deviation. "Numerical Experiment and Analysis" evaluates the proposed method. "Conclusions" summarizes the work of this article and tells the shortcomings.

## LITERATURE REVIEW

Group decision making is the participation of multiple people and multiple subjects in forming a consensus of preferences about an uncertain decision problem and making a choice about the alternatives. The purpose of group decision-making (GDM) is to make full use of the experience and wisdom of multiple decision-makers (DMs), give full advantage of different knowledge structures, and make the decision-making results more objective and close to reality (*Ding et al., 2020*; *Wu et al., 2020*; *Tang & Liao, 2021*). Generally, GDM with more than 20 DMs (*Pan et al., 2020*; *Liu et al., 2021b*) is divided into LSGDM. Due to the complexity of the decision-making process, the solution to the LSGDM will take a lot of time and money. The clustering process is indispensable for the LSGDM (*Pan et al., 2020*; *Zhong & Xu, 2020*; *Ding et al., 2020*; *Du et al., 2020*, *2021*; *Liu, Zou & Wu, 2020*; *Liu et al., 2021a*; *Wu & Liao, 2021*). In recent years, the rapidly growing rise of electronic democracy has led to a dramatic increase in the complexity and ambiguity of the decision-making process. It is common for DMs to provide hesitant and fuzzy decision information. The study of clustering with hesitant and fuzzy decision information has gradually become the favored object of scholars.

*Ma et al. (2019)* studied the LSGDM clustering problem and proposed a clustering method of multistage hesitant fuzzy language sets. Firstly, the similarity measurement of DMs is proposed by setting the expected distance and hesitation similarity. Secondly, the fuzzy equivalent clustering method is applied to divide the DMs'clusters. The main problem is that the expected distance setting requires reference experience, and there is no standard measurement. In addition, the time complexity of calculating the distance is higher. *Lin et al. (2018)* carried out clustering research on the language term set of hesitation probability and proposed an order-first clustering method based on the similarity of ideal values. *Li (2021)* proposes a trust-adaptive-based clustering model for fuzzy preference relationships to divide decision-makers into internal and external groups, and then implement different consensus processes for different classified DMs. Given the flexibility of fuzzy language, *Liu, Huang & Qiu (2022)* establishes a clustering and consensus model with fuzzy value judgment based on fuzzy number quantization. *Meng (2017)* proposed a new concept of multiplicative consistency for intuitionistic fuzzy

preference relations (IFPRs) and constructed a 0–1 hybrid clustering method to judge the consistency of IFPRs. In fact, the most effective clustering method applied to fuzzy sets is the fuzzy C-means (FCM) algorithm. The main feature of the algorithm is that the clustering centers are determined in advance, and the clustering results are easily affected by the centers. In reality, it is difficult to specify the required clustering centers in FCM. In addition, due to DMs' experience, knowledge, and understanding of the evaluation object, most of the decision information is incomplete. Meanwhile, incomplete decision information cannot be directly clustered. Therefore, LSGDM clustering research with incomplete decision information has gradually received the attention of scholars.

*Tian, Nie & Wang (2019)* proposed to use of the social network analysis (SNA) method to manage the relationship among DMs and solve LSGDM problems. Firstly, a multi-objective model of trust relationships among DMs is constructed based on a social trust network. Then the missing values in the incomplete interval type-2 fuzzy sets (IIT2FSs) decision matrix are estimated. Considering the knowledge coverage of experts, *Liu et al. (2021a)* proposed to estimate the trust relationship among experts based on the knowledge coverage trust propagation operator. *Lu et al. (2022)* proposes a social network clustering algorithm based on the grey model for incomplete fuzzy relations to manage distrust behavior. *Chu et al. (2020)* proposed a fuzzy clustering method based on the social network for community division. First, community similarity and centrality are used to measure the importance of community division, and then the incomplete fuzzy preference relation is supplemented. *Khameneh, Kilicman & Md Ali (2022)* constructed an attribute-based clustering method based on fuzzy multiple graphs in structure and attribute similarity. In view of the incomplete information about the effect of interpersonal relationships on group decision-making, *Wu et al. (2019)* proposes to use the type two language trust function to simulate the trust relationship among DMs. First, the ordered weighted average (OWA) is used to obtain the complete trust relationship. And then the missing information in the decision matrix is estimated.

Although there are a few studies on clustering after supplementing decision information, there are still the following problems:

(1) When building a social trust network (STN), only part of the interactive information is used to spread the trust relationship, and the trust relationship is equal (*Li et al., 2022*). Single consideration of explicit trust or implicit trust. However, it is obvious that explicit trust and implicit trust play different roles in the trusted network. In addition, the existing research does not consider the inhibition effect of distrust relationship and trust decay, which makes the supplementary trust information deviate from reality and reduces the quality of decision-making.

(2) Most of the existing studies only consider the relay trust nodes among DMs but ignore the fact that the preference of DMs reflects the similarity to a certain extent (*Du et al., 2020*). It is also an important supplement to the trust relationship and plays an important role in supplementing decision information. Currently, the decision information supplement methods that have been proposed are based on aggregation operators or improvements based on them, which have the problem of information loss (*Labella et al., 2019*; *Zheng et al., 2021*; *Kumar & Chen, 2022*).

(3) Existing researches on clustering only rely on preference similarity or trust relationships (*Ma et al., 2019*; *Pan et al., 2020*; *Zhong & Xu, 2020*; *Sahu et al., 2020*; *Zheng et al., 2021*; *Mandal, Samanta & Pal, 2022*). Otherwise, in practical problems, preference and trust play different roles in the clustering process. In addition, most of the clustering research is based on the improvement of the FCM algorithm. This algorithm needs to determine the clustering center in advance, which makes the clustering result easily affected. It is highly subjective that there is no uniform standard for centroid selection, and manually specifying the required centroids is difficult. Moreover, the algorithm is unstable and susceptible to noise interference. Besides, the existing hierarchical hesitant fuzzy clustering algorithm doesn't use the information of the data set itself to determine the weights of the distance function.

Based on the above analysis, a novelty incomplete hesitant fuzzy information supplement and clustering method for LSGDM are proposed. The main contributions are as follows:

(1) A new STN construction method is proposed, which constructs the global and local trust network, and fuses explicit and implicit trust. In the process of trust propagation, a multiplication function is applied to simulate the decay of trust value, and a threshold is used to simulate the inhibition of the distrust relationship.

(2) Extend the collaborative filtering algorithm to situations with hesitating fuzzy variables, after fusing trust-preference use collaborative filtering to recommend neighbors. A new decision information supplement function is proposed, which considers the willingness of DMs and the reference significance of close neighbors. The method of fusing trust and preference makes the decision information supplement more scientific and improves the decision quality.

(3) A weighted hesitant fuzzy clustering algorithm based on the relative standard deviation is proposed. Firstly, a method to complement the set of hesitant fuzzy elements is given, and a new weight distance function based on the relative standard deviation theory is given. Secondly, cluster centers are selected using density peaks to optimize the selection and avoid the existence of subjective factors in the manual selection. Finally, clustering is performed to reduce the scaling of LSGDM problems.

## PRELIMINARIES

This section reviews some concepts related to Hesitation Fuzzy Sets and Social Trust Network analysis, meanwhile giving some method-related definitions.

### Hesitant fuzzy set

Since people usually cannot use an accurate number to measure the degree of the description membership in the characteristics of things, quantify the degree of membership and give the class to which a certain characteristic of a thing belongs. *Torra (2010)* firstly defined the hesitant fuzzy set (HFS), which is defined as follows:

**Definition 1** Let $M = \{\mu_1, \mu_2, \ldots, \mu_k\}$ be a set containing k membership functions, then the HFS $h_M$ is defined as Eq (1):

$$h_M(x) = \{\mu_{1(x)}, \mu_{2(x)}, ..., \mu_{k(x)})\} \tag{1}$$

where $x \in X$ obtains the HFS $M$ under the mapping of $\mu$, subsequently, *Chen, Xu & Xia (2014)* then used precise mathematical symbols to represent HFS, defined as follows the Eq. (2):

$$H = \{<x, hA(x)> | x \in X\} \tag{2}$$

where $h_A(x)$ represents the set of all possible affiliations $x$ belonging to the $H$. Their values are real numbers between [0,1]. For convenience, $h_A(x)$ r is called a hesitating fuzzy element (HFE), which is the basic unit of HFS. HFE sets the lower bound to $h^-(x) = \min h(x)$, the upper to $h^+(x) = \max h(x)$.

To be more suitable for LSGDM problem clustering and subsequent calculation, we redefine the concepts of HFE and hesitating degree and give the corresponding formulas.

**Definition 2** Let $H(X) = \{h(x_1), h(x_2), ..., h(x_d)\}$ be a hesitant set of the data object $X$, which contains d attributes. $h(x)$ is the HFE set, that is, through the function $h$, $x_j(j = 1, 2, ..., d)$ the data object $X$ is mapped to a membership set, and $h(x_j) \in [0, 1]$. For example, $h(x_1)$ represents the HFE about the first attribute feature of the data object $X$, and $H(X)$ is a set of HFE about $X$.

To calculate the distance between HFSs, the hesitation degree was defined as $\lambda x = |h(x)|$, which represents the number of membership values in fuzzy elements of the attribute $x$, and ensures $\lambda x_{aj} = \lambda x_{bj}$. $l(H(X_i)) = \sum_{j=1}^{d} \lambda_{x_{ij}} (j = 1, 2, ..., d)$ can also be used to represent the sum of hesitancy degrees of all attributes. That $l(H(X_a)) = l(H(X_b))$ must be guaranteed when calculating the distance of the HFS. When $l(H(X_a)) < l(H(X_b))$ calculating the hesitation degree of each attribute HFSs $\lambda_{x_{ij}}$. If $\lambda_{x_{aj}} < \lambda_{x_{bj}}$, then add $\lambda_{x_{bj}} - \lambda_{x_{aj}}$ membership values $H(x_{aj})$ to ensure $\lambda_{x_{aj}} = \lambda_{x_{bj}}$.

References in this article give the conversion relationship between linguistic variables and HFSs (*Ren, 2019*; *Ma et al., 2019*), as shown in Table 1.

## Social network analysis

Social network analysis (SNA) mainly studies the relationships among social entities, including centrality, prestige, and trust relationships (*Wu et al., 2019*; *Labella et al., 2019*; *Du et al., 2020*; *Tang & Liao, 2021*). There are three traditional expressions of trust relationship in SNA as shown in Table 2.

Sociometric: Relationship data usually appears in two forms 0 or 1. 0 means that there is no direct trust relationship and 1 means that there is a direct trust relationship.

Graph: The network is considered a graph composed of nodes connected by lines.

Algebraic: The advantage of this representation is that it allows us to distinguish several different relations and represent combinations of relations.

The clustering problem existing research based on trust relationships usually constructs a STN according to SNA. The constructed trust network consists of group nodes $V = \{e_1, e_2, ..., e_q\}$ and edges $L = \{l_1, l_2, ..., l_n\}$, in which nodes represent individuals or organizations and edges represent the relationship among nodes. The trust network representation method only has direct trust relationships and can't represent the trust

**Table 1 Linguistic terms and their corresponding HFS.**

| Linguistic terms | HFS |
|---|---|
| Very low (VL) | [0, 0, 0, 0.05; 0.9] |
| Low (L) | [0.05, 0.12, 0.18, 0.25; 0.9] |
| Medium low (ML) | [0.2, 0.32, 0.38, 0.45; 0.9] |
| Medium (M) | [0.4, 0.52, 0.58, 0.65; 0.9] |
| Medium high (MH) | [0.6, 0.72, 0.78, 0.85; 0.9] |
| High (H) | [0.75, 0.82, 0.88, 0.95; 0.9] |
| Very high (VH) | [0.95, 1, 1, 1; 0.9] |

**Table 2 Different representation schemes in social network analysis.**

| Sociometric | Graph | Algebraic |
|---|---|---|
| $A = \begin{bmatrix} 0 & 1 & 1 & 1 & 1 & 0 \\ 0 & 0 & 0 & 0 & 1 & 0 \\ 0 & 1 & 0 & 0 & 0 & 0 \\ 0 & 0 & 1 & 0 & 1 & 1 \\ 0 & 0 & 1 & 0 & 0 & 1 \\ 0 & 0 & 1 & 0 & 0 & 0 \end{bmatrix}$ |  | $\begin{array}{ll} E_1RE_2 & E_1RE_3 \\ E_1RE_4 & E_1RE_5 \\ E_2RE_5 & E_3RE_2 \\ E_4RE_3 & E_4RE_5 \\ E_4RE_6 & E_5RE_3 \\ E_5RE_6 & E_6RE_3 \end{array}$ |

strength among DMs. According to the generalization and summary of trust relationships in sociology, trust relationship among entities is transitive, and the trust relationship can propagate freely among nodes under the restriction of constraint rules. As shown in Fig. 1, an example of trust relationship propagation is illustrated. In the figure, the nodes set V constitutes the DMs set, and the directed edge set among the corresponding nodes constitutes the trust relationship set. The weights on the edges in the figure represent the trust values among DMs. Then the directed graph G = (V, E) is the trust network propagation.

Combined with the sociological communication theory and considering the strong and weak connection relationship in the communication path among nodes in the STN, this article makes the following assumption in the process of trust relationship transmission: trust relationships can spread in the weak connection communication path, but there will be a loss, which leads to the attenuation of trust strength, and the attenuation amplitude is proportional to the length of the weak connection communication path. Based on this assumption, this article proposes two rules that must be obeyed in the propagation process of trust weak connection paths of trust relationship:

Rule (1) of trust relationship propagation:

$$\forall(Dir\_Trust(a,b), Dir\_Dir(b,c) \in [0,1]) \Rightarrow Indir\_Trust(a,c) \in [0,1]$$

Rule (2) of trust relationship propagation:

$$\forall(Dir\_Trust(a,b), Dir\_Dir(b,c) \in [0,1]) \Rightarrow Indir\_Trust(a,c)$$
$$< \min(Dir\_Trust(a,c), Dir\_Trust(b,c))$$

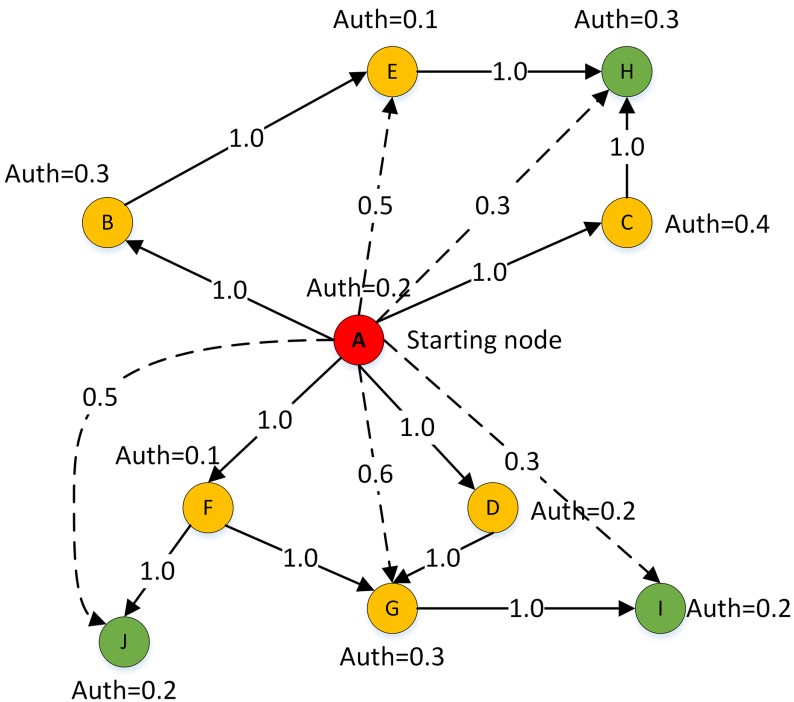

**Figure 1  Trust relationship propagation.** 

where $Dir\_Trust(a, b)$ indicates that there is an explicit trust relationship among DMs. $Indir\_Trust(a, c)$ indicates that DMs can obtain an implicit trust relationship through a weak connection path.

## THE PROPOSED METHOD FOR THE LSGDM CLUSTERING PROBLEM

In this article, we propose a fusion trust-preference decision information supplement method and a weighted hesitant fuzzy clustering algorithm based on relative standard deviation. The developed model consisting of ninth steps is visualized in Fig. 2.

The framework of the supplementary method of fusion trust-preference decision information is introduced in "Fusion Trust-preference Decision Information Supplement Method". The weighted hesitant fuzzy clustering algorithm based on the relative standard deviation is introduced in "Weighted Hesitant Fuzzy Clustering Algorithm Based on Relative Standard Deviation".

### Fusion trust-preference decision information supplement method

The fusion trust-preference decision-making information supplement method is mainly composed of two parts: trust network construction and the decision-making information supplement method. The details are shown in Fig. 3.

As shown in Fig. 3, DMs provide trust relationships and incomplete decision information according to the LSGDM problem description. Firstly, we construct the trust network, including constructing the global trust network and local trust network through trust propagation. In this process, the attenuation of trust value and the inhibitory effect of

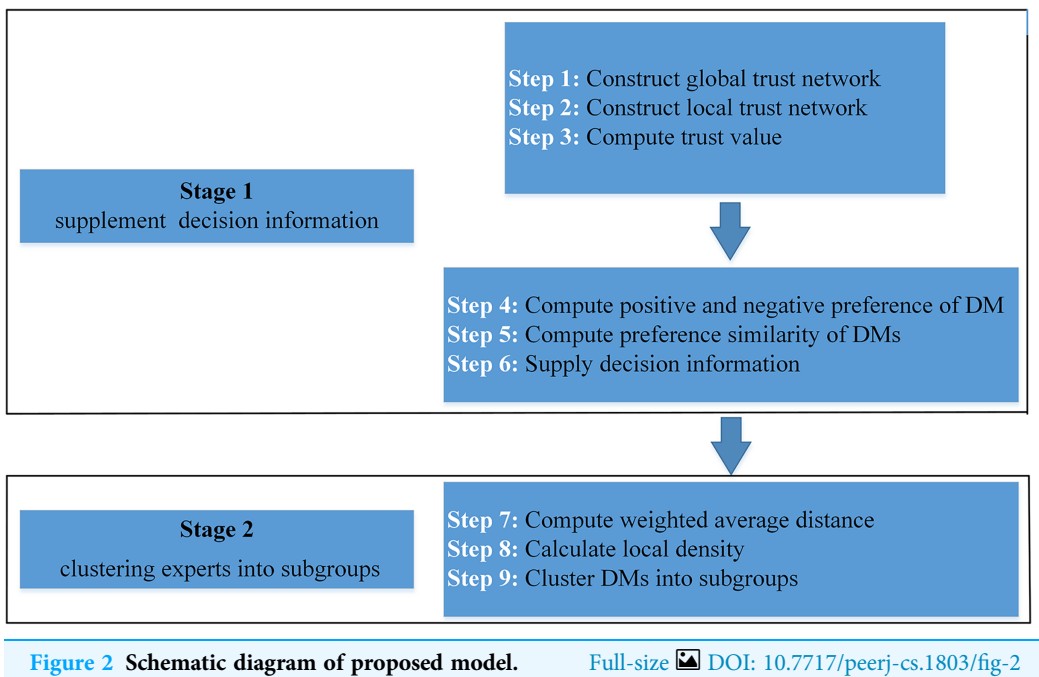

**Figure 2 Schematic diagram of proposed model.**     

distrust on propagation are considered. Secondly, we need to fuse trust preference. Finally, the appropriate DMs' neighbors are recommended by the collaborative filtering algorithm, and the missing decision information is supplemented.

### Social trust network construction

Based on sociology and graph theory, this article considers the loss of trust value and the inhibiting effect of distrust relationship of trust relationship. Figure 4 shows the process of trust network construction.

As shown in Fig. 4, there are three steps in the STN construction process:

(1) Initialize the global trust network based on the list of direct trust relationships among DMs, and obtain the global trust relationship of each DM.

(2) By detecting the propagation path of possible weak connections, mining new trust relationships, and enriching the implicit trust relationships. In this process, the multiplication function is used to simulate the attenuation of trust value in the actual trust propagation process, and the threshold is used to simulate the inhibition effect of distrust.

(3) Fusing global and local trust relationships and updating the trust list of DMs.

#### Step 1: Construct global trust network

According to the trust relationship list of DMs, the initial operation is used to establish a global trust network. The result is shown in Fig. 5.

In Fig. 5, nodes represent DMs, and directed edges indicate that there is a direct trust relationship among DMs.

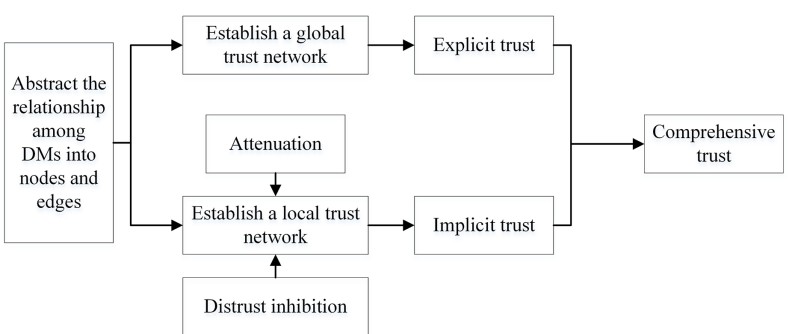

Figure 3 **The overall architecture of decision information supplementation method.**

Figure 4 **The construction of the social trust network.**

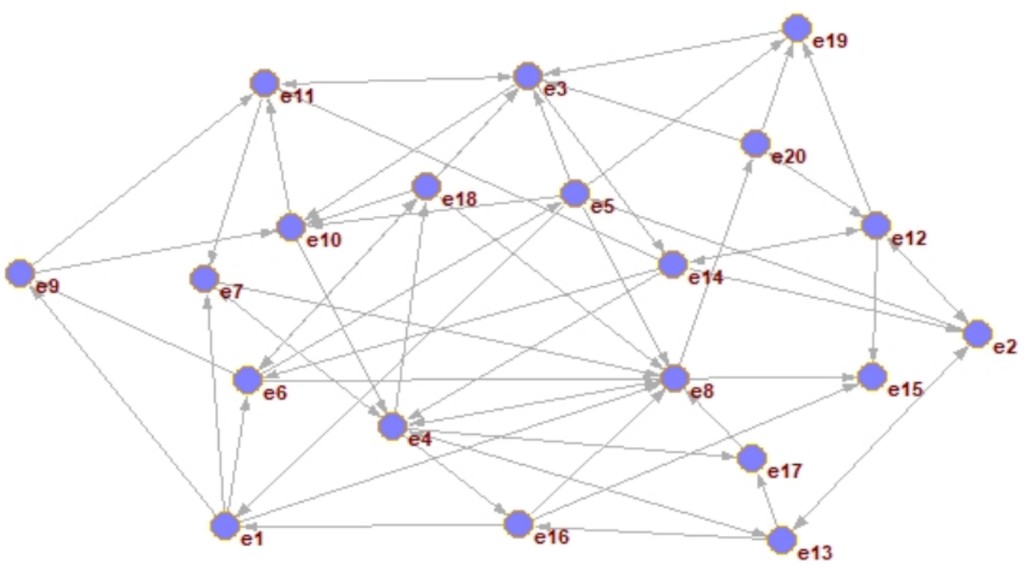

**Figure 5 The construction of the global trust network.**

The explicit trust degree of the DM in the global scope of the directed network graph G = (V, E) is calculated. The parameters for weight in this section mainly come from the in-degree information of the DM.

The in-degree information of the target DM is a good indicator of how much other DMs in the network trust him, and the trust relationship is divided into positive and negative. Therefore, when using the DM's trust degree information as the calculation parameter to obtain the explicit trust degree, it is necessary to distinguish the positive and negative trust relations. The explicit trust degree calculation is shown in Eq. (3):

$$Dir\_trust(A) = \frac{(Indg_+(G, A) - Min(Indg_+(G, *))) - (Indg_-(G, A) - Min(Indg_-(G, *)))}{Max(Indg_{+/-}(G, *)) - Min(Indg_{+/-}(G, *))} \quad (3)$$

$Indg_+(G, A)$ indicates that DM A obtains positive trust evaluation information from other DMs in the global trust network. $Indg_-(G, A)$ represents that DM A obtains negative trust evaluation information. $Max(indg_{+/-}(G, *))$ and $Min(indg_{+/-}(G, *))$ respectively represents the maximum in-degree and minimum in-degree for the DM to obtain trust evaluation in the directed graph G. The larger the explicit trust degree obtained by Eq. (3), the more direct the trust relationship between the DM and others. It means that the DM may be the core node, and the probability of being concerned by others is also high. If the positive trust information is less than the negative trust information, the DMs' explicit trust degree in the global trust network is negative.

**Step 2: Construct local trust network**

Although initializing operations can obtain global trust, a single initialization cannot solve the sparse trust relationship. To solve this problem, corresponding rules must be formulated for trust relationship propagation to enrich and expand the trust relationship in the local trust network.

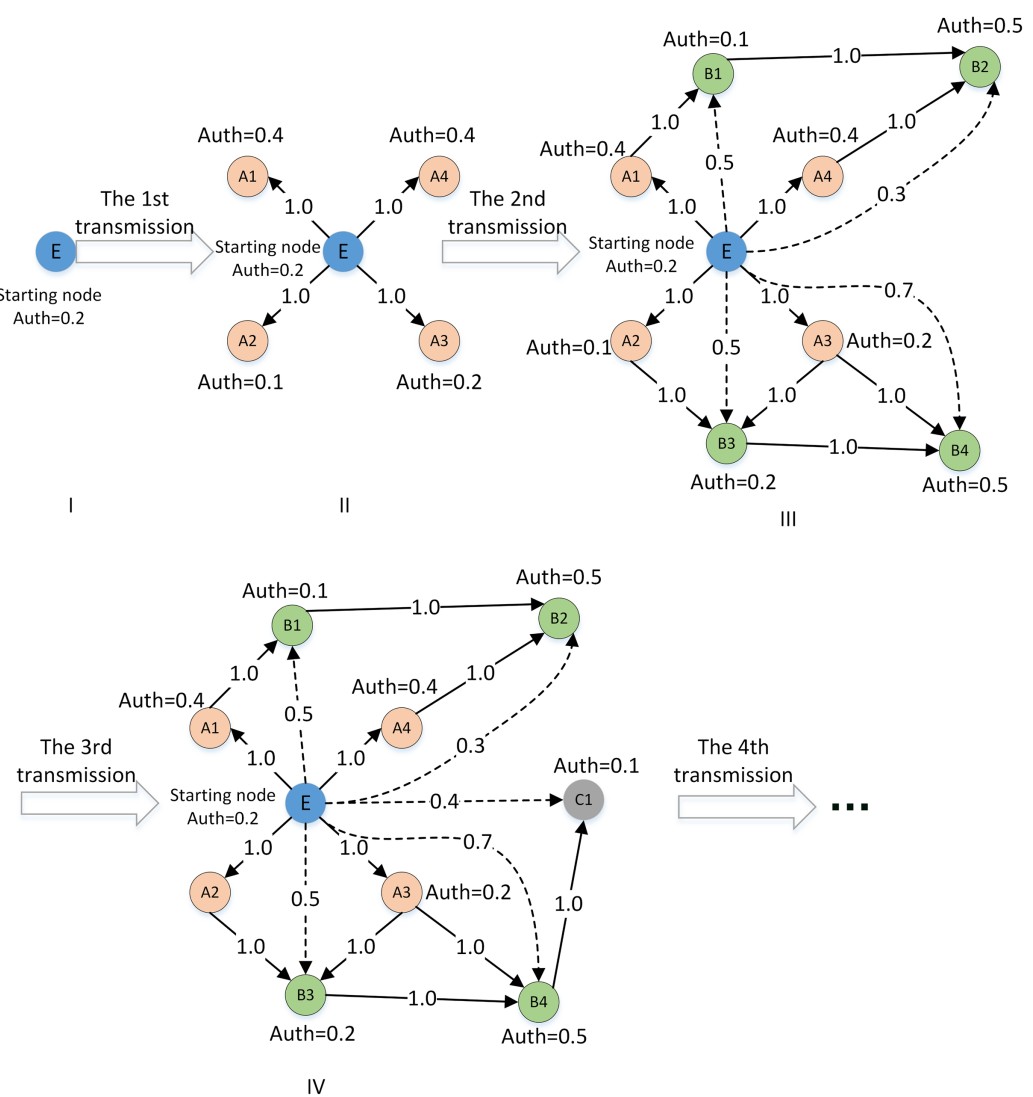

**Figure 6 Example of local trust network path detection.**

Therefore, a weakly connected path detection method of the local trust network is designed in this article. In this method, DMs are self-centered and the length of the weakly connected path is taken as the radius to construct the local trust relationship of DMs. The extended example is shown in Fig. 6. To detect new trust relationships in the local network centered on DM E. The solid line between nodes indicates that there is a strong connection path among DMs, while the dashed line indicates that there is a weak connection path.

Combined with the trust relationship propagation rule (1) and rule (2), the multiplication function is introduced to calculate the implicit one-way trust relationship between decision makers on the weakly connected path. At the same time, the trust weights of other DMs with implicit trust relationships on the weakly connected path of the central DM are calculated. The calculation of the trust weight is as follows:

(1) Selecting the DM with a reachable weak connectivity path and its connectivity path

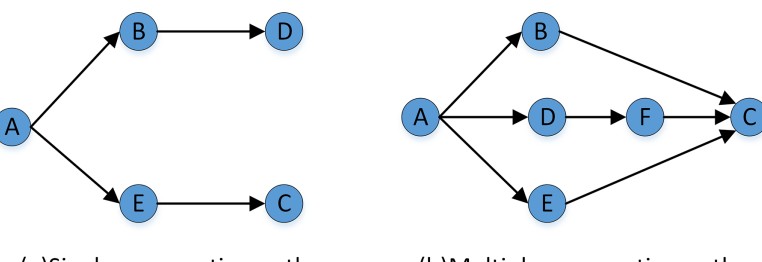

(a)Single propagation path      (b)Multiple propagation paths

**Figure 7 Examples of trust relationship propagation in the connected path.**

Each DM in the global trust network is taken as the central starting node, and the local trust network is extended as shown in Fig. 6. Because there are positive and negative trusts, the trust propagation process also examines the positive and negative trust relationships. If a node in the propagation path has too many negative trust ratings, the DM is not suitable as a relay node for trust propagation. The detection process of this weak connection path should be ended in advance. Therefore, in the process of selecting the reachable weak connection path, the trust evaluation of negative information of the relay node must be less than or equal to the threshold value $\sigma$. The threshold is set as shown in Eq. (4):

$$\sigma = \frac{1}{3} \times \frac{Indg - (G, A)}{Max(indg + / - (G, *))} \tag{4}$$

$Indg_-(G, A)$ represents the negative trust evaluation information obtained by DM$A$. $Max\big(indg_{+/-}(G, *)\big)$ represents the maximum in-degree for the DM to obtain trust evaluation in the directed graph G. Considering the propagation path of the weak connection is too long, it may cause noise interference to the new trust relationship. We set the maximum propagation distance as $Max(Path\_len) = 6$.

(2) Calculate the implicit trust value among DMs under the single weak connection path

If there is an accessible weakly connected path between DM C and initial center DM A in the network, namely $path = (a, t_1, t_2, \ldots, t_m, c)$. Although the node chain $t_1, t_2, \ldots, t_m$ as relay nodes ensures a weak connection between the beginning and end nodes, as shown in Fig. 7A. The multiplication (Eq. (5)) can be used to calculate the implicit trust relationship weight.

$$
Indir\_trust(a, c) = \frac{1}{Max(Indg + / - (G, *))^m} \times
$$
$$
\left( \frac{\sum Dir\_Trust(*, a)}{N(TrustList(a))} \times \frac{\sum Dir\_Trust(*, t1)}{N(TrustList(t1))} \times \cdots \times \frac{\sum Dir\_Trust(*, tm)}{N(TrustList(tm))} \right) \tag{5}
$$

$\sum Dir\_Trust(*, t_1)$ represents the explicit trust relationship that DM$^{t1}$ in the network obtains from others. $\frac{\sum Dir\_Trust(*, t_1)}{N(TrustList(t_1))}$ means that after summation of each DM's explicit trust in degree value, it is evenly distributed its trust resource to the DMs with associated trust out-degree. $N(TrustList(t_1))$ indicates the scaling of the DM's trust list, $Max\big(Indg_{+/-}(G, *)\big)$ represents the maximum in-degree for the DM to obtain trust

evaluation in the directed graph G. Its function is mainly to ensure that the trust relationship complies with the range constraints during the propagation process.

(3) Calculate the implicit trust value of DMs under multiple weak connection paths

If there are multiple weak connection paths with different lengths or parallel between the initial DM and the target DM, as shown in Fig. 7B. The implicit trust relationship between the DMs needs to be fused, and Eq. (6) is used to fuse and calculate the average implicit trust value from the central node to the target DM.

$$\overline{Indir\_trust(a,c)} = \frac{1}{|path(a,c)|} \sum\nolimits_{path(a,c)} Indir\_Trust(a,c) \tag{6}$$

where $path(a,c)$ represents the weakly connected and reachable path between DM A and DM C. $|Path(a,c)|$ denotes the number of weakly connected reachable paths.

**Step 3: Compute trust value**

In the social trust network, the directed edges (A, C) indicate that there is a trust relationship between DM A and DM C. The trust relationship is affected by two factors. On the one hand, DM A should consider the overall trust evaluation of C in the global network as a reference for the trust influence of DM C. On the other hand, DM A needs to examine the strength of the connection between himself and C based on his trust relationship list. Therefore, the value of the trust relationship between DMs will be calculated in combination with the above two factors, and is shown in Eq. (7):

$$Trust(a,c) = \begin{cases} \xi \times Dir\_Trust(a,c) + (1-\xi) \times Auth(c) \\ \xi \times Indir\_Trust(a,c) + (1-\xi) \times Auth(c) \end{cases} \tag{7}$$

where $Trust(a,c)$ is the value of the trust relationship between A and C. $Dir\_Trust(a,c)$ indicates the explicit trust value between the DM A and DM C. $Indir\_Trust(a,c)$ represents the implicit trust value on the weakly connected path. $Auth(c)$ is the total global trust value of DM C in the STN. $\xi$ is the proportion coefficient, which belongs to $[0,1]$. Its role is to adjust the explicit trust value and implicit trust value.

### Preference similarity (PS) of DMs

Trust models in social networks reflect only the "historic" behavior of experts. In order to estimate missing values more accurately, this article proposes the PS for the expert's "current" behavior (decision matrix) according to the characteristics of the LSGDM problem.

**Step 4: Compute positive and negative preference**

In the LSGDM, the DM can give the approval level of the alternative based on his understanding of the attribute. Through the evaluation of the specific attribute given by the DMs, the preference of the DMs can be further judged, and the similarity among DMs can be explored. This is of great significance for selecting suitable DMs to supplement the missing items. This article studies the incomplete hesitant fuzzy preference matrix given by DMs. For an attribute of the alternative, if DM gives a higher average score than others, then his preference for the attribute will be enhanced. Calculate the degree of preference by

studying the preference information of the DMs' attributes. The positive preference $Pref_{At}^p$ and negative preference $Pref_{At}^n$ of DM A for attribute $t$ are Eqs. (8) and (9) respectively:

$$Pref_{At}^p = \frac{1}{|I_A^p(t)|} \times \sum_{i \in I_A^p(t)} \frac{R_{Ai} - \overline{R}}{\sqrt{\sum_{j-1}^{I} R_{Aj}^2}} \tag{8}$$

$$Pref_{At}^n = \frac{1}{|I_A^n(t)|} \times \sum_{i \in I_A^n(t)} \frac{\overline{R} - R_{Ai}}{\sqrt{\sum_{j-1}^{I} R_{Aj}^2}} \tag{9}$$

According to the positive preference and the negative preference degree, calculating the preference degree of the DM A for the attribute $t$ is Eq. (10):

$$Pref_A(t) = Pref_{At}^p - Pref_{At}^n \tag{10}$$

**Step 5: Compute preference similarity of DMs**

If $PrefA(t) > 0$ indicates that DM A has a positive preference for the attribute $t$. On the contrary, $PrefA(t) < 0$ indicates that DM A has no preference for the attribute $t$. The preference of different DMs for attributes can indirectly reflect the similarity. Therefore, the PS among DMs is calculated based on the result of the preferences for attributes calculated by Eq. (11):

$$PS(A, C) = \frac{\sum_{t \in (T_A \cap T_C)} Pref_A(t) \times Pref_C(t)}{\sqrt{\sum_{t \in T_A} Pref_A^2(t)} \times \sqrt{\sum_{t \in T_C} Pref_C^2(t)}} \tag{11}$$

$A_t$ and $C_t$ represent the attribute set of alternatives that have been graded by DM A and DM C respectively.

**Step 6: Supply decision information**

In this article, the collaborative filtering recommendation algorithm is introduced into the LSGDM problem. Based on the traditional collaborative filtering recommendation algorithm, a collaborative filtering algorithm using DMs' trust and preference combination strategy to replace a single factor is proposed. The algorithm architecture is shown in Fig. 8. The DMs' trust matrix and preference similarity matrix are used as the input, and the recommended nearest neighbors are used as the output.

To solve the problem of decision deviation caused by the lack of decision information, fusion trust and preference use the collaborative filtering algorithm to find suitable neighbors for the target DMs. In the implementation of the algorithm, we need to consider the trust relationship and the DMs' preference factors. New hybrid weight is obtained by introducing the weighted harmonic function, and the comprehensive similarity is used for near neighbor selection.

To reflect the joint effect of trust and preference in the selection of recommending the most similar DMs. In the article, the preference degree and the trust value are reconciled before selecting the nearest neighbors.

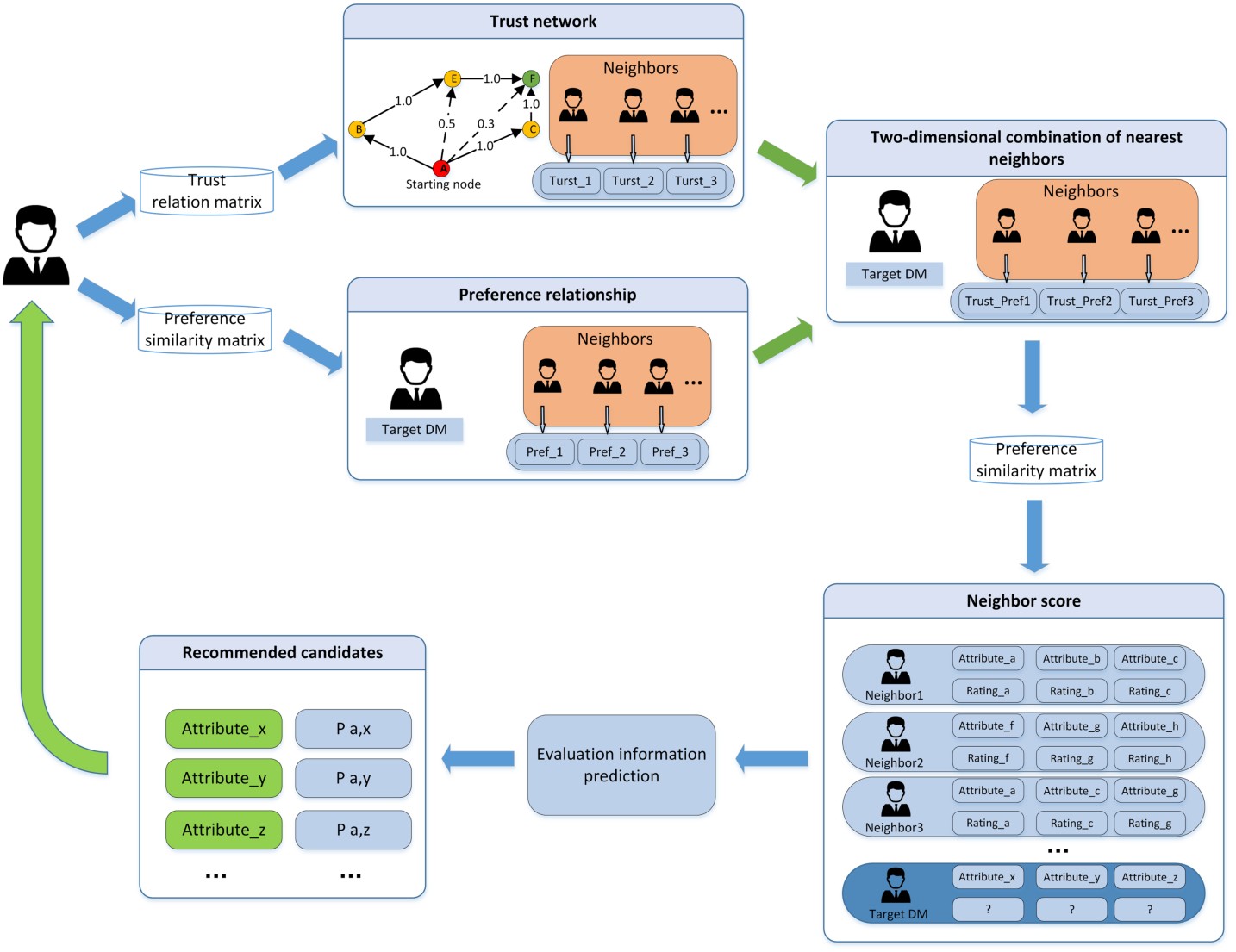

**Figure 8  The structure of collaborative filtering algorithm.**

Considering that the preference and trust factors are equally important, this article adopts the two factors value equal allocation strategy. The comprehensive similarity value and the harmonic average among DMs are shown in Eq. (12):

$$Trust\_PS(a,c) = 2 \times \frac{PS(a,c) \times Trust(a,c)}{PS(a,c) + Trust(a,c)} \tag{12}$$

After calculating and ranking the comprehensive similarity matrix. Firstly, the two-dimensional similar nearest neighbors of the target scale are selected according to the sorting rules. Secondly, the neighbor evaluation data is provided as input to the evaluation prediction model, and the recommendation candidates and recommendation values are calculated. The prediction method fuses two factors: the opinions of the DM and the

---

**Algorithm 1** Decision-making information prediction method.

**Input:** $T n \times n$, $Pref n \times n$, $HFSs$

**Output:** $Complete\_HFSs$

1. *for i in range* $(1, n)$
2.    *for j in range* $(1, n)$
3.       $Pref_p(i, j) = \dfrac{\sum\limits_{t \in T} Pref_i(t) \times Pref_j(t)}{\sqrt{\sum\limits_{t \in T} Pref_i^2(t)} \times \sqrt{\sum\limits_{t \in T} Pref_j^2(t)}}$   //Pref between $DMi$ and $DMj$
4.       $Trust\_Pref(i, j) = mixBothfactors(T(i, j), Pref(i, j))$ //fusing dual-factor
5.    *end*
6. *end*
7. $Trust\_pref = sort(Trust\_pref)$ // Descending sort
8. *for i in range* $(1, n)$
9.    *if* $isIncomplete(HFSs_i)$// Expert evaluation matrix is incomplete
10.       $neigbors = selectNeigbors(Trust\_Pref)$ // Select nearest neighbors
11.       $Complete\_HFSs_i = fillValue(HFSs, neigbors)$ // Fill in missing values
12.    *end*
13.    $Complete\_HFSs_i = HFSsi$
14. *end*
15. *return* $Complete\_HFSs_i$

---

reference significance of the nearest neighbors. The prediction of weighted decision information is shown in Eq. (13).

$$Pa, i = \overline{ra} + \frac{\sum_{c \in Sa} (Rc, i - \overline{rc}) \times Trust\_PS(a, c)}{\sum_{c \in Sa} |Trust\_PS(a, c)|} \tag{13}$$

where $P_{a,i}$ indicates the prediction evaluation value. $S_a$ is the near neighbors set of DM A, $R_{c,i}$ is the value in the comprehensive similarity matrix. $\overline{r_a}$ is the mean value of decision evaluation for a certain attribute of all DMs. The specific implementation process is shown in Algorithm 1.

## Weighted hesitant fuzzy clustering algorithm based on relative standard deviation

In practical problems, due to factors such as their own experience, knowledge, and understanding of the evaluation object, each DM gives different degrees of hesitation for different attributes. When performing cluster analysis on decision-making problems, traditional fuzzy clustering cannot effectively solve decision-making problems in practical scenarios. Therefore, some scholars have proposed a clustering algorithm for hesitant fuzzy sets. The existing fuzzy clustering algorithm does not use the information of the dataset itself to determine the weight of the distance function, and the computational complexity and space complexity of the cluster center are both exponential, which is not

suitable for LSGDM scenarios. Aiming at the above problems, a weighted hesitant fuzzy clustering algorithm based on the relative standard deviation is proposed. Firstly, the method of complementing the hesitant fuzzy element set is given, and a new calculation formula of distance function weight is given based on the coefficient of variation theory. Then the cluster center is selected by using the density peak. It not only reduces the computational complexity of the cluster center but also improves the adaptability to data sets of different scales and arbitrary shapes. Besides the time and space complexity of the algorithm are also reduced to the polynomial level.

After the discussion in the previous section, we get the complete decision matrix, and then use it for clustering. Considering that in practical problems, decision makers give different degrees of hesitation for different attributes. Therefore, before the implementation of the clustering algorithm, it is necessary to check and fill the HFE set of all data objects to ensure that the hesitant degree of each attribute feature is consistent. However, Using risk preference to increase the membership value in the original set of HFE may bias the results to DM and cause the decision evaluation of others to lose value (*Maihama, Zandi & Naderi, 2019*). Considering the above problems, this article adopts the following method of supplementing HFEs.

**Step 7: Calculate weighted average distance**

**Definition 3** Let $h(x_a) = \left\{ \gamma_{a_{j1}}, \gamma_{a_{j2}}, ..., \gamma_{a_{jn}} \right\}$ and $h(x_{b_j}) = \left\{ \gamma_{b_{j1}}, \gamma_{b_{j2}}, ..., \gamma_{b_{jn}} \right\}$ be the set of HFEs of the attribute $j$ of data objects A and B. If $\lambda x_{a_j} < \lambda x_{b_j}$, using Eq. (14) to calculate the mean value $\overline{\gamma'}$ of $h(x_{a_j})$, and add $\overline{\gamma'}$ to expand $h(x_{a_j})$ to make $\lambda x_{a_j} = \lambda x_{b_j}$, where the mean value can be expressed as Eq. (14):

$$\overline{\gamma'}_j = \frac{\sum\limits_{k=1}^{\omega} \gamma a j_k}{\omega} \tag{14}$$

Combining the Hamming distance, this article adopts the weighted hesitant Hamming distance. The value is shown in Eq. (15):

$$dham(A, B) = \sum_{j=1}^{d} \omega j \left[ \frac{1}{\lambda} \sum_{\sigma=1}^{\lambda_j} |h\sigma(xaj) - h\sigma(xbj)| \right] \tag{15}$$

In addition, the DM can set the weight manually according to the importance of each attribute of the data object. The average weights $\omega = (1/d, 1/d, ..., 1/d)^T$ can be set without specific requirements (*Wu et al., 2020*). This article adopts a new weight calculation method based on the data information and the relative deviation theory. Firstly, the mean value $\overline{\gamma'}$ and standard deviation $S_j$ of all membership degrees of the attribute $j$ in the data set are calculated, and they can be calculated by Eqs. (16) and (17):

$$\overline{\gamma_j} = \frac{\sum\limits_{i=1}^{n} \sum\limits_{k=1}^{\lambda_{xij}} \gamma ij_k}{\sum\limits_{i=1}^{n} \lambda x_{ij}} \tag{16}$$

$$Sj = \left[\frac{\sum\limits_{i=1}^{n} \sum\limits_{k=1}^{\lambda x_{ij}} (\gamma ij_k - \overline{\gamma_j})^2}{n}\right]^{\frac{1}{2}} \tag{17}$$

$$RSD_j = \frac{S_j}{\overline{\gamma_j}} \tag{18}$$

Bring the calculated results of $\overline{\gamma_j}$ and $S_j$ into Eq. (18), which is the relative deviation theory. If the degree of variation of the attribute is greater, it means that the membership value of the attribute is more unstable, and the DM has more differences about the attribute. Therefore, it is necessary to reduce the weight of the attribute. Take the reciprocal of $RSD_j$ to get $RSD_j'$ as shown in Eq. (19), and finally $\omega_j$ is obtained by Eq. (20).

$$RSD_j' = \frac{1}{RSD_j} \tag{19}$$

$$\omega_j = \frac{RSD_j'}{\sum\limits_{j=1}^{d} RSD_j'} \tag{20}$$

**Step 8: Calculate local density**

**Definition 4** (*Capuano et al., 2018*) Let the data object of data set S be $X_i$ and its local density be $\rho_i$, which can indicate the number of data objects whose distance from $X_i$ is less than $d_c$. Equations (21) and (22) are as follows:

$$\rho_i = \sum_{j \neq i} \chi(d_{ij} - d_c) \tag{21}$$

$$\chi(x) = \begin{cases} 0 & x \geq 0 \\ 1 & x < 0 \end{cases} \tag{22}$$

When the amount of data is large, reduce the probability of equal local density values of data objects within the data set. Gaussian kernel formula is used to replace Eqs. (21), (23) and (24) are as follows:

$$\rho_i = \sum_{j \neq i} \exp\left(-\left(\frac{d_{ij}}{d_c}\right)^2\right) \tag{23}$$

$$\rho_i = \sum_{j \in KNN} \exp(-d_{ij}) \tag{24}$$

where $d_{ij}$ is the distance between the HFS of the data object $X_i$ and $X_j$. The value of the

truncation distance $d_c$ usually makes the truncation distance contain $1\% \sim 2\%$ the data sample size. When the data set sample size is small, there may be no data points in $d_c$, thus affecting the calculation result of the local density $\rho_i$.

**Step 9: Cluster DMs into subgroups**

Through the improvement of Eq. (24), the nearest neighbor K can be selected as $1\% \sim 2\%$ the data sample size to reduce the influence on the local density (*Xie et al., 2016*).

$$\delta_i = \begin{cases} \min_{j:\rho_i < \rho_j}(d_{ij}), & \exists j \rho_i < \rho_j \\ \max_j(d_{ij}), & otherwise \end{cases}, j = 1, 2, ..., n \tag{25}$$

From Eq. (25), when the data object has the largest local density, its relative distance $\delta_i$ is the distance from the furthest away from it. The relative distance $\delta_i$ of other data objects represents the distance between the closest object and the local density greater than $X_i$. The cluster centers are selected by calculating the value $\tau_i = \rho_i \times \delta_i$. Finally, the data objects are allocated to the clusters that are closest to the center and the whose local density is greater than that of the data objects. According to the above idea, the basic steps of the algorithm are shown in Algorithm 2.

# NUMERICAL EXPERIMENT AND ANALYSIS

The weighted hesitant fuzzy clustering algorithm based on relative standard deviation and the fusion trust-preference choice information supplement approach will be examined from three angles in this part. An example demonstrating the method's viability is provided in "Illustrative Example", with data sourced from *Wang et al. (2018)* and *Tian, Nie & Wang (2019)*. Comparing the suggested supplementary technique with the OWA operator (*Lu et al., 2022*) and *Tian, Nie & Wang*'s *(2019)* supplemental method are the two key comparisons with other approaches made in "Comparison with Other Methods" The suggested clustering algorithm and *Tian, Nie & Wang*'s *(2019)* clustering method are compared.

## Illustrative example

A sixth of China's total land area is made up of the Xinjiang Uygur Autonomous Region, which is situated in northwest China. Because of its location in the heart of the Eurasian continent, Xin serves as a vital logistical hub and the focal point of China's 2013 proposal for the "Belt and Road Initiative (BRI)". The building of the logistic park in Xin is crucial to the BRI. Xin plans to select an optimal location among four locations to construct a logistics park. The locations are Altai (a1), Hami (a2), Korla (a3), and Kashgar (a4). 20 experts, who are denoted by $V_e = \{e_1, e_2, ..., e_{20}\}$ participating in the evaluation decision for potential locations based on three factors. The factors are social (c1), environmental (c2), and economic (c3). The experts provide their evaluations for the locations with respect to each factor using linguistic terms. The decision information is presented in Appendix 1, where $\varphi$ indicates missing values. The linguistic trust evaluations among

**Algorithm 2** Weighted hesitant fuzzy clustering algorithm based on relative standard deviation.

**Input:** *HFSs_matrix*

**Output:** *Clusters of DMs*

1. *A = GetAttribute (HFSs)* //Get all Attributes of evaluation matrix
2. *for i in range A*
3. $\omega_i = \dfrac{RSD_j}{\sum_{j=1}^{d} RSDj}$ //Computer attribute weight
4. *end*
5. *for i in range HFSs*
6.    *for j in range A*
7.       $\lambda^j = getHesitation\left(HFSs_{ij}\right)$ // Get the hesitation $\lambda^j$ of attribute j
8.       $\overline{\gamma j} = \dfrac{\sum_{k=1}^{\omega} \gamma HFSsijk}{\omega}$ // complete the HFEs according to Eq. (15)
9.    *end*
10. *end*
11. $dham(A, B) = \sum_{j=1}^{d} \omega j [\dfrac{1}{\lambda} \sum_{\sigma=1}^{\lambda_j} |h\sigma(xaj) - h\sigma(xbj)|]$// Calculate the distance between data objects according to Eq. (16)
12. *for i in range DMs*
13.    $\rho_i = \sum_{j \neq i} \chi\left(d_{ij} - d_c\right)$
14.    *s.*
15.    $\tau i = \rho i \times \delta i$
16. *end*
17. $C = getCenter(\tau)$ //select the cluster centers by sorting the $\tau$ value.
18. *for i in range DMs*
19.    $c = selectCluster(\tau, C, dham)$
20.    $DM_i \in clusters(c)$ // Data objects are placed in the cluster
21. *end*
22. *return* clusters

experts are presented in Appendix 2. The linguistic ratings are then transformed into HFSs by referring to Table 1.

### *Supplementary results*

To obtain the best cluster (s), the following steps are undertaken:

Step 1: Fig. 5 presents the results of building the global trust network based on the explicit relationship. The explicit trust degree of each expert is calculated according to Eq. (3). Take DM $e_3$ as an example, explicit trust relationships including $e_5 \rightarrow e_3$, $e_{11} \rightarrow e_3$, $e_{18} \rightarrow e_3$, $e_{20} \rightarrow e_3$. Where $e_{20} \rightarrow e_3$ is negative trust, others are positive trust. The global direct explicit trust of DMs is calculated as in Eq. (3):

$$Dir\_trust(e3) = \frac{(Indg + (G, e3) - Min(indg + (G, *))) - (Indg - (G, e3) - Min(indg - (G, *)))}{Max(indg + / - (G, *)) - Min(indg + / - (G, *))}$$

$$= \frac{(4 - 2) - (1 - 0)}{7 - 2} = 0.2$$

Step 2: Construct a local trust network based on "Social Trust Network Construction". Take the trust propagation from DM 1 to DM 3 as an example. By referring to the method (*Tian, Nie & Wang, 2019*) omitting the trust path with a length greater than or equal to four, there are two trust paths, which are $e_1 \rightarrow e_8 \rightarrow e_{20} \rightarrow e_3$ and $e_1 \rightarrow e_9 \rightarrow e_{11} \rightarrow e_3$. Compared with *Tian, Nie & Wang (2019)*, the paths obtained are less $e_1 \rightarrow e_6 \rightarrow e_5 \rightarrow e_3$ and $e_1 \rightarrow e_6 \rightarrow e_{18} \rightarrow e_3$. This is because this article considers the inhibition effect and the attenuation of trust transmission, which is more in line with the characteristics of trust relationship propagation in reality.

Step 3: The DMs and their connected paths on the existence of reachable weakly connected paths are chosen in accordance with "Social Trust Network Construction". The implicit trust values of single or multiple weakly connected paths are calculated based on Eqs. (5) and (6). The trust propagation from $e_1$ to $e_3$ can be taken as an example:

$$\overline{Indir\_trust(e1, e3)} = \frac{1}{|path(e1, e3)|} \sum_{path(e1,e3)} Indir\_trust(e1, e3)$$

$$= \frac{1}{|path(e1, e3)|} \left( \frac{1}{(Max(Indg + / - (G, *)))^2} \left( \frac{\sum Dir\_trust(*, e1)}{N(TrustList(e1))} \times \frac{\sum Dir\_trust(*, e8)}{N(TrustList(e8))} \times \frac{\sum Dir\_trust(*, e20)}{N(TrustList(e20))} \right) \right.$$

$$+ \frac{1}{(Max(Indg + / - (G, *)))^2} \left( \frac{\sum Dir\_trust(*, e1)}{N(TrustList(e1))} \times \frac{\sum Dir\_trust(*, e9)}{N(TrustList(e9))} \times \frac{\sum Dir\_trust(*, e11)}{N(TrustList(e11))} \right) \right)$$

$$= \frac{1}{|path(e1, e3)|} \left( \frac{1}{(Max(Indg + / - (G, *)))^2} \times \right.$$

$$\left. \left( \frac{\sum Dir\_trust(*, e1)}{N(TrustList(e1))} \times \frac{\sum Dir\_trust(*, e8)}{N(TrustList(e8))} \times \frac{\sum Dir\_trust(*, e20)}{N(TrustList(e20))} + \frac{\sum Dir\_trust(*, e9)}{N(TrustList(e9))} \times \frac{\sum Dir\_trust(*, e11)}{N(TrustList(e11))} \right) \right)$$

The implicit trust value $\overline{Indir\_trust(e_1, e_3)} = 0.4564$ is obtained by substituting the calculation results of each stage. The comprehensive trust value is calculated according to Eq. (7) in which the proportion coefficient $\xi$ of explicit trust value and implicit trust value can be adjusted according to different scenarios. In this article, we consider explicit trust and implicit trust to be equally important, so we take the value of 0.5.

$$Trust(e_1, e_3) = \xi \times Indir\_Trust(e_1, e_3) + (1 - \xi)Auth(e_3) = 0.3283$$

Step 4: Positive and negative preference degrees are calculated according to Eqs. (8) and (9) in "Preference Similarity (PS) of DMs". The final preference degree of DM A for attribute t is obtained through Eq. (10). The preference similarity among DMs is determined according to Eq. (11).

Taking DM $e_1$ and DM $e_3$ as examples, their linguistic evaluation matrices are obtained from Appendix 1. According to the transformation rules in Table 1, we obtain HFS preference matrices for $e_1$ and $e_3$. The columns represent attributes and the rows represent alternatives.

$$HFSe1 = \begin{bmatrix} \{0.6, 0.72, 0.78, 0.85; 0.9\} & & \{0.2, 0.32, 0.38, 0.45; 0.9\} \\ \{0.4, 0.52, 0.58, 0.65; 0.9\} & \{0.6, 0.72, 0.78, 0.85; 0.9\} & \\ \{0.6, 0.72, 0.78, 0.85; 0.9\} & & \{0.4, 0.52, 0.58, 0.65; 0.9\} \\ \{0.75, 0.82, 0.88, 0.95; 0.9\} & \{0.4, 0.52, 0.58, 0.65; 0.9\} & \{0.6, 0.72, 0.78, 0.85; 0.9\} \end{bmatrix}$$

$$HFSe1 = \begin{bmatrix} \{0.6, 0.72, 0.78, 0.85; 0.9\} & & \{0.2, 0.32, 0.38, 0.45; 0.9\} \\ \{0.6, 0.72, 0.78, 0.85; 0.9\} & & \{0.4, 0.52, 0.58, 0.65; 0.9\} \\ \{0.4, 0.52, 0.58, 0.65; 0.9\} & \{0.75, 0.82, 0.88, 0.95; 0.9\} & \{0.6, 0.72, 0.78, 0.85; 0.9\} \\ \{0.75, 0.82, 0.88, 0.95; 0.9\} & \{0.4, 0.52, 0.58, 0.65; 0.9\} & \{0.4, 0.52, 0.58, 0.65; 0.9\} \end{bmatrix}$$

As this is solely for statistical preference, the trend rather than a particular value is what matters. To facilitate the calculation, the score function is introduced as shown in Eq. (26) to simplify the preference matrix (*Rodríguez et al., 2014*).

$$S(h) = \frac{\sum_{j=1}^{l(h)} \delta(j)\gamma_j}{\sum_{j=1}^{l(h)} \delta(j)} \tag{26}$$

where $h$ denotes the HFE, $l(h)$ is the number of elements in $h$. $\delta(j)$ is the positive monotonically increasing order of subscript $j$. The simplified preference matrix is as follows:

$$HFSe1 = \begin{bmatrix} 0.82 & & 0.55 \\ 0.69 & 0.82 & \\ 0.82 & & 0.69 \\ 0.89 & 0.69 & 0.82 \end{bmatrix} \qquad HFSe3 = \begin{bmatrix} 0.82 & & 0.55 \\ 0.82 & & 0.69 \\ 0.69 & 0.89 & 0.82 \\ 0.89 & 0.69 & 0.69 \end{bmatrix}$$

By substituting the simplified matrix into Eqs. (8)–(10). It is calculated that the preference of DMs $e_1, e_3$ for each attribute is and $Prefe_3 = \{-0.43, 0.29, 0.43, 0.36\}$. Finally, according to Eq. (11) to obtain $PS(e_1, e_3) = 0.91$.

Step 5: replacing the single component in the collaborative filtering algorithm with the trust and preference factors combined from DM. The target DM who has the strongest trust relationship and the closest preference is chosen to fill in the DM's missing value. The algorithm architecture is shown in Fig. 8. For the complete preference matrix, please see the Supplemental Materials.

### Clustering results

To separate DMs into distinct clusters, the weighted hesitation fuzzy clustering algorithm based on relative standard deviation is supplemented with the entire preference matrix. It mainly includes: adding missing hesitation fuzzy elements according to Eq. (14), calculating the weighted hesitation fuzzy distance according to Eq. (15), calculating the

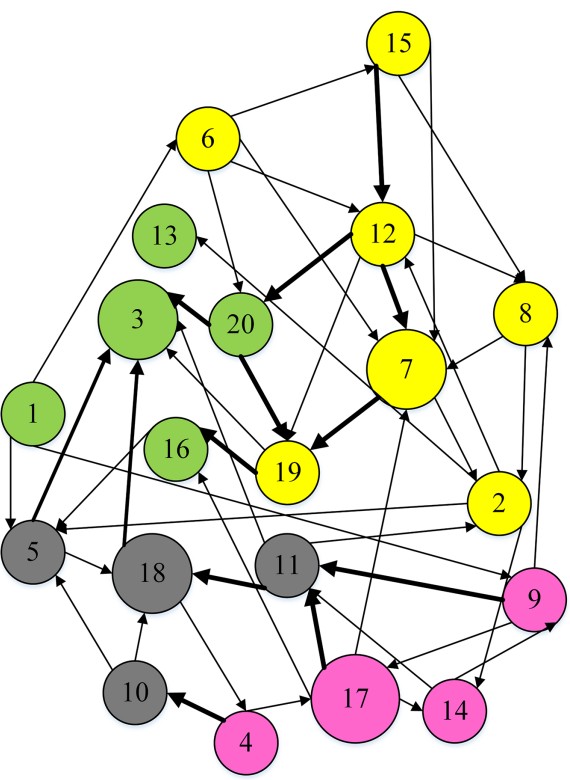

**Figure 9 Clustering result.** 

distance weight according to Eq. (20), and calculating the local density according to Eq. (21), Calculate the selected cluster center $\tau_i$ according to Eqs. (24) and (25). In this data set $t_3 = 0.6979$, $t_{17} = 0.6296$, $t_7 = 0.5903$, $t_{18} = 0.5683$, is more than $t_8 = 0.3368$. The final is $t_3 > t_{17} > t_7 > t_{18} > t_8 > \cdots > t_{15} > t_{16}$. The DMs are divided into 4 clusters, including $V1 = \{e_3, e_1, e_{13}, e_{20}, e_{16}\}$, $V2 = \{e_7, e_8, e_{12}, e_6, e_2, e_{19}, e_{15}\}$, $V3 = \{e_9, e_{14}, e_{17}, e_4$, $V4 = \{e_5, e_{11}, e_{18}, e_{10}\}$, as shown in Fig. 9.

## Comparison with other methods

### Comparison of decision information supplement methods

The purpose of the decision information supplement is to bring the supplemental results as close to DM preferences as possible while also improving the consistency of clustering results. As a result, clustering results can be used to validate the usefulness of supplemental outcomes. The consistency degree is the most essential metric of the clustering algorithm's efficacy for LSGDM situations. The fusion trust preference decision information supplement approach is evaluated against Zhang's supplement method (*Tian, Nie & Wang, 2019*) and the widely utilized OWA supplement method (*Lu et al., 2022*) to determine its efficacy for the LSGDM problem. Three average consensus degree indicators are created to evaluate the method's validity, inspired by some earlier research (*Xu, Du & Chen, 2015*; *Zheng et al., 2021*).

    To ensure the objectivity of the experimental results, the experimental process adopts the same original decision information and the weighted hesitant fuzzy clustering method

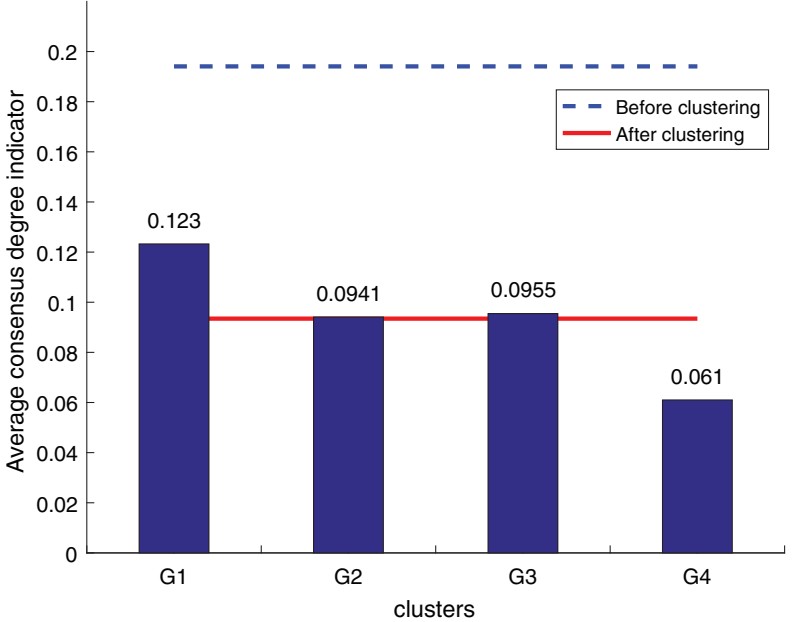

**Figure 10 OWA operator supplements consistency level index results.**

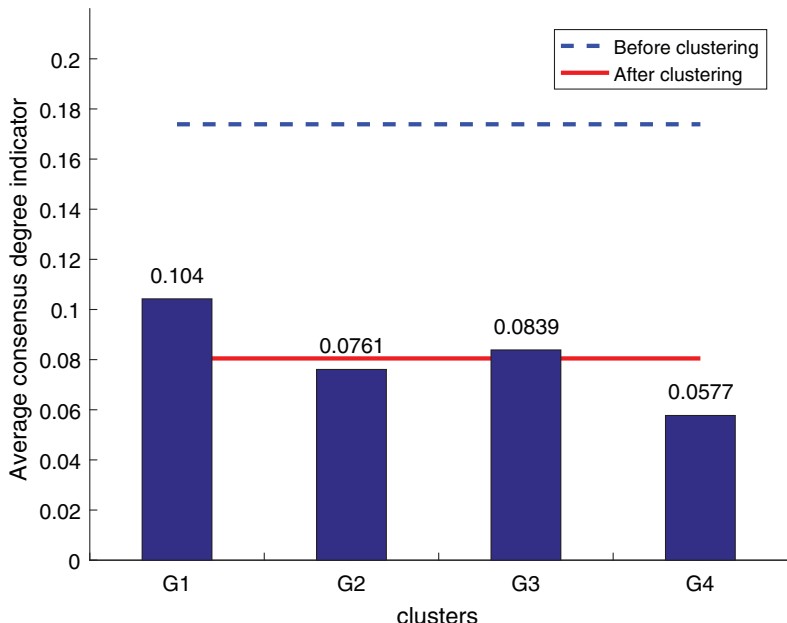

**Figure 11 Zhang's supplementary method (*Tian, Nie & Wang, 2019*) consistency level index results.**

based on the relative standard deviation. According to the three consistency indexes, the smaller the value, the higher the degree of consistency, and the method is more effective in solving the LSGDM. The results of the three indexes are shown in Figs. 10 to 12:

According to Figs. 10 to 12, it can be seen that the average consistency level of the overall cluster is 0.067, which is much smaller than the pre-clustering average consistency level of

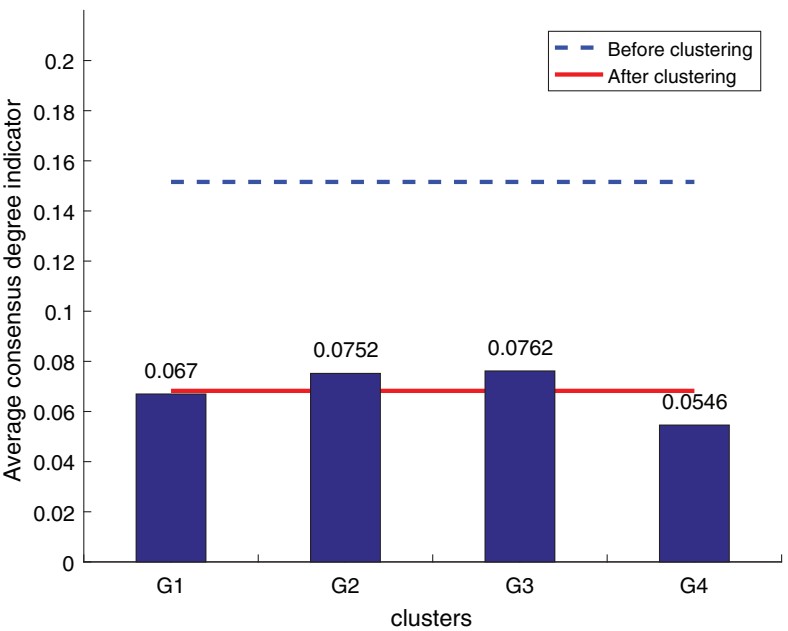

**Figure 12 Our method consistency level index results.**

0.1568. Meanwhile, it is smaller than the OWA algorithm 0.0955 and Zhang's supplementary method 0.08. After clustering, the four clusters' average consistency level is likewise noticeably higher than that of other supplemental techniques. The degree of consistency increases with a smaller consistency index. Consequently, the computation results demonstrate the applicability of the suggested fusion trust-preference decision information supplement method for the LSGDM problem.

When dealing with LSGDM problems involving incomplete hesitation fuzzy information, the fusion trust and preference decision information supplement method works well. The following are the causes:

In order to make the established trust network more realistic and scientific, we first simulate the actual trust decay and the inhibitory effect of distrust relationships on the propagation of trust relationships. Secondly, taking into account the various roles that preferences and trust relationships have in complementing. To suggest neighbors, we combine trust and preference and apply a collaborative filtering algorithm. Lastly, we also care about the reference significance of the closest neighbors with the most similar preferences who are also the most trustworthy and similar to the DM, taking into account the DM's own wishes. As a result, the complementary outcomes match reality better.

### Comparison of clustering algorithm

Given that the consistency level is the most crucial metric for assessing how well the clustering algorithm performs on the LSGDM problem. Thus, the consistency level comparison is used in this article to demonstrate the benefits of the clustering method.

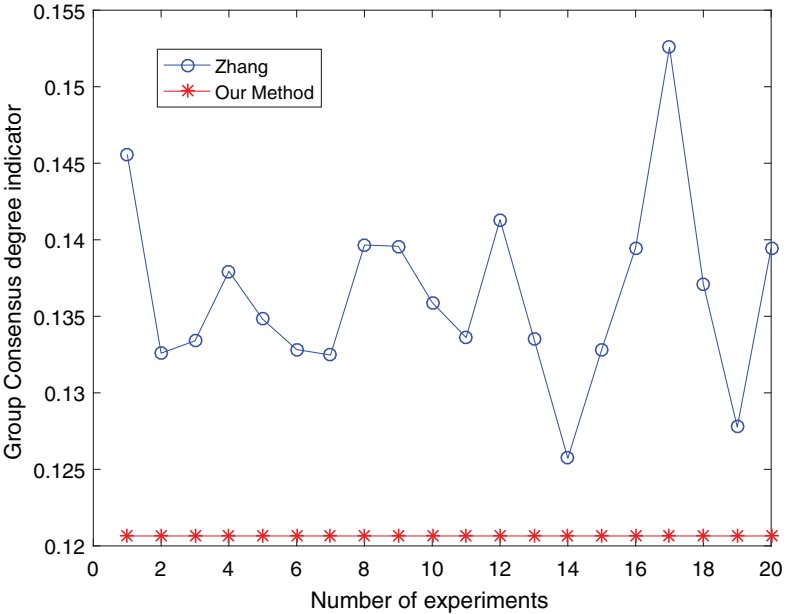

**Figure 13 Comparison of group consensus for cluster number = 3.**

The decision matrix for clustering comparison is supplemented with the fusion trust-preference decision information prediction method to guarantee the experiment's objectivity and viability.

The number of clusters is taken from three to five and clustered 20 times respectively to make the experimental results objective to overcome the effects of the number of clusters and random. The comparative results are shown in Figs. 13 to 15:

From Figs. 13 to 15, it is obvious:

When the number of clusters = 3, the group average consistency level after clustering by Zhang's method fluctuates from 0.125 to 0.152 with a large range, while the proposed method is stable at 0.12. when the number of clusters=4, the group average consistency level of the proposed method is 0.07, and Zhang's method fluctuates from 0.085 to 0.12. When the number of clusters = 5, the group average consistency level of our clustering method is 0.093, and Zhang's method fluctuates from 0.1 to 0.13. The smaller the group averages consistency level indicator, the higher the degree of consistency and the better the clustering results (*Xu, Du & Chen, 2015*; *Zheng et al., 2021*), it can be seen that our clustering method is more effective than the Zhang's method for LSGDM regardless of the number of clusters.

The automatic center selection of our clustering algorithm is effective because, according to the experimental results, the overall cluster consistency level is lowest and the effect is greatest when there are four clusters. Furthermore, our clustering method is more stable and robust because, irrespective of the number of clusters, the overall cluster consistency level tends to be a straight line.

When compared to Zhang's method, the hesitant fuzzy clustering method based on relative standard deviation has the aforementioned benefits. The following are the causes:

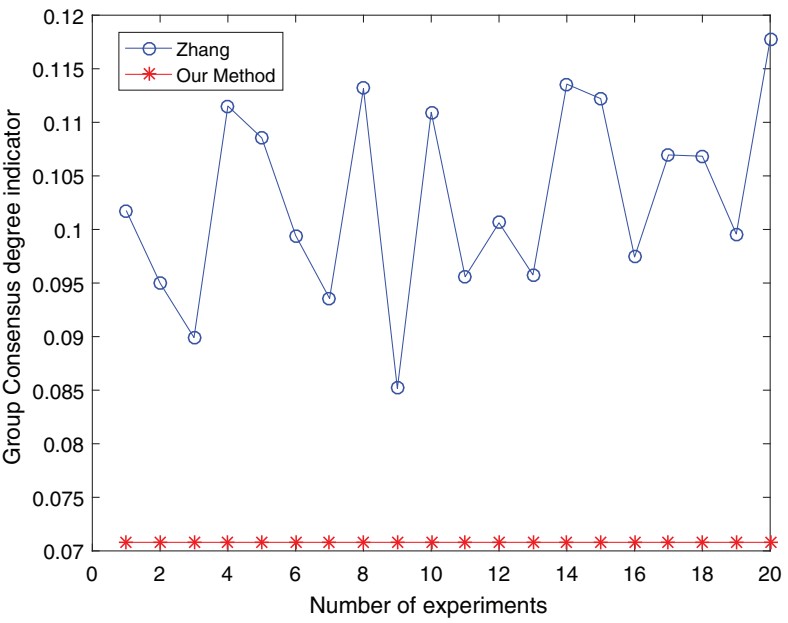

**Figure 14 Comparison of group consensus for cluster number = 4.**

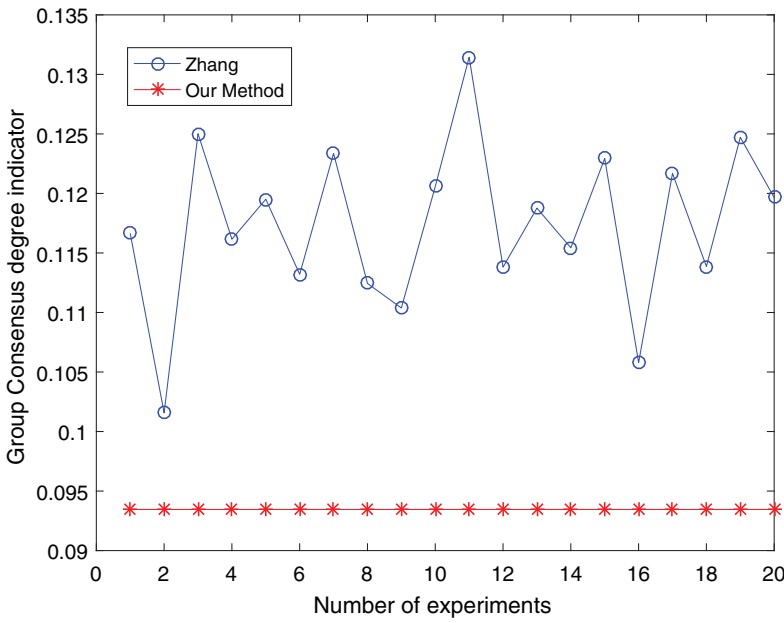

**Figure 15 Comparison of group consensus for cluster number = 5.**

First, the clustering process takes into account both preference and trust factors, grouping decision makers who are more similar into a single class. Second, when figuring out the distance, the relative standard deviation theory is presented. The weight of the distance function is determined by looking at the data set itself. In order to prevent noise interference and local optima, choose the cluster center as efficiently as possible. As a

result, there is an improvement in intra-cluster consistency and the accuracy of the clustering results.

## CONCLUSIONS

Aiming at the clustering problem of large-scale group decision-making with incomplete hesitant fuzzy information, this article proposes a fusion trust-preference decision information supplementary method and a weighted hesitant fuzzy clustering algorithm based on relative standard deviation. According to the experimental results, the following conclusions can be drawn:

(1) The supplementary results are guaranteed by the fusion trust-preference decision-making information supplement method presented in this article. In the meantime, a trustworthy network that reflects DM reality is formed. Furthermore, the accuracy and consistency of supplementary results are enhanced by taking into account the supplementary meaning of neighbors and the DMs' own volition.

(2) The relative standard deviation based weighted hesitation fuzzy clustering algorithm is used to classify DMs. Secondly, relative standard deviation and weighted hesitant fuzzy distance are introduced to solve the problems of the high computational complexity of cluster centers and unreasonable distance weights. Furthermore, cluster centers are chosen automatically to prevent interference from noise. Furthermore, the degree of intra-cluster consistency following clustering is efficiently measured by the clustering distance, which takes into account both preference and trust factors.

Our approach successfully addresses the clustering problem in Large-Scale Group Decision Making (LSGDM) with incomplete, hesitant, and fuzzy information, yielding favorable outcomes. However, the model developed in this study does not take into account the overlapping phenomenon among decision-makers across clusters or the influence of decision-makers from overlapping communities on clustering results. Additionally, our research lacks a thorough consideration of scalability and throughput analysis. Furthermore, the model is specifically designed for LSGDM problems with incomplete hesitant fuzzy decision information and does not consider constructions in complex scenarios, such as heterogeneous information. In future research, we will give due consideration to these issues and, in terms of baseline comparisons, select more relevant studies (*Tang et al., 2019*; *Tian et al., 2021*).

### Funding
The authors received no funding for this work.

### Competing Interests
The authors declare that they have no competing interests.

### Author Contributions
- Jingdong Wang conceived and designed the experiments, prepared figures and/or tables, and approved the final draft.

- Wenhui Wang conceived and designed the experiments, performed the experiments, performed the computation work, prepared figures and/or tables, and approved the final draft.
- Fanqi Meng performed the experiments, prepared figures and/or tables, authored or reviewed drafts of the article, and approved the final draft.
- Peifang Wang conceived and designed the experiments, performed the experiments, authored or reviewed drafts of the article, and approved the final draft.
- Xuesong Wang analyzed the data, performed the computation work, prepared figures and/or tables, and approved the final draft.
- Shuang Wei analyzed the data, authored or reviewed drafts of the article, and approved the final draft.
- Tong Liu performed the experiments, authored or reviewed drafts of the article, and approved the final draft.
- Shuaisong Yang performed the computation work, prepared figures and/or tables, and approved the final draft.

## Data Availability

The experimental data are available in the Supplemental Files.

## Supplemental Information

Supplemental information for this article can be found online at http://dx.doi.org/10.7717/peerj-cs.1803#supplemental-information.

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
