# Peer review of "A novel incomplete hesitant fuzzy information supplement and clustering method for large-scale group decision-making"

_PeerJ Computer Science, doi:10.7717/peerj-cs.1803_

## Round 0.1 · original submission · Major Revisions

The authors should prepare a major revision based on reviewer comments.

Reviewer 1 ·

Basic reporting

1. There are a few unclear sentences that can be rephrased for better clarity of ideas by the intended audience. for example, in the abstract sections, the opening sentence can be used as
"Clustering is an effective means to reduce the scaling of large-scale group decision-making (LSGDM)". Authors are requested to proofread the manuscript for such issues to enhance the impact of the proposed study.

2. In the abstract section, the problem statement and motivation for the proposed solution are either too vague or too broad to devise the need for such systems. For example, there have been numerous works proposed by several researchers to improve the fairness and transparency in distance calculation, and clustering centers for clustering approaches. Authors are requested to provide a narrow roadmap for the problem or challenges this study is trying to resolve.

3. The introduction section is immature and way of its domain. Authors have tried to provide comparative analysis between the studies but these comparisons are usually suitable in the review section. The introduction section serves as a foundation for the entire research paper, guiding readers into the study, explaining its significance, and preparing them for the content that follows. It should be clear, concise, and well-structured to effectively convey the purpose and context of the research.

4. There is no literature review section in the submission copy. Authors are required to provide a review section as it plays a crucial role in building the intellectual framework for the research paper and highlighting the research's contribution to the field. It should be well-organized, comprehensive, and focused on the specific research questions or objectives of the study.

Experimental design

The idea of clustering is quite robust and old for the problem domain. The authors are advised to provide a clear breakdown of the contributions made by this study. For example, the following studies are quite relevant to the methods used:

https://dl.acm.org/doi/10.1016/j.inffus.2019.02.001
https://www.sciencedirect.com/science/article/abs/pii/S1568494619307549

Given this, only one new formula for calculation would fall into slight contribution.

Validity of the findings

The results provided by the authors are not sufficient to provide a clear and thorough model evaluation. For example, the what-if analysis of model behavior of different studies would have added more impact to the solution.

Similarly, the scalability and throughput analysis is also important for the clustering algorithms. Similarly, for the baseline comparisons more relevant studies could have been selected. Few are shared in the above comments.

In addition to these, lastly., I would add a slight note to add more descriptive information about the case and assumptions that are being leveraged by the proposed study. The submitted manuscript have not provided any such information about the data.

Rest is all fine.

Reviewer 2 ·

Basic reporting

The paper addresses an important issue in the context of large-scale group decision-making (LSGDM) by proposing a novel approach that combines incomplete, hesitant fuzzy information supplementation with clustering. It outlines the challenges associated with traditional clustering methods in LSGDM, such as the lack of effective decision-making information supplementation, the high computational complexity of cluster centers, and inappropriate distance weights. The proposed approach integrates trust degradation, trust propagation, and network building to optimize cluster center selection.

Experimental design

The clustering process appears to be well-structured and systematic, leading to the formation of four distinct clusters of decision-makers.

Validity of the findings

The proposed method achieves good results in solving the clustering problem associated with incomplete, hesitant, and ambiguous information in LSGDM.

---

## Round 0.2 · Minor Revisions

The authors have revised the article based on comments. The reviewer has provided feedback to revise the article for another round.

Reviewer 1 ·

Basic reporting

The authors have done good work revising the document. However, the introduction section should provide the paper section summary at the end. Plus the knowledge gap for the problem is still missing in the section. Please revise the section to justify the need for the problem by highlighting the problems and gaps in the recent research work.

Experimental design

NA

Validity of the findings

The results provided by the authors are not sufficient to provide a clear and thorough model evaluation. For example, the what-if analysis of model behavior of different studies would have added more impact to the solution.

Similarly, the scalability and throughput analysis is also important for the clustering algorithms. Similarly, for the baseline comparisons more relevant studies could have been selected. Few are shared in the above comments.

Add more baselines for the experimental comparison.

Additional comments

NA

---

## Round 0.3 · accepted · Accept

The article is accepted after two rounds of revision and based on reviewer comments.

Reviewer 1 ·

Basic reporting

No further comments.

Experimental design

NA

Validity of the findings

Still can be improved for better representation of ideas. More focus to make findings more interesting. Write short sentences with a clear message.